# Benchmarking the predictive capability of human gait simulations

**Maarten Afschrift**[1]*, **Dinant Kistemaker**[1], **Maarten Bobbert**[1], **Friedl De Groote**[2]

**1** Faculty of Behavioural and Movement Sciences, Vrije Universiteit Amsterdam, Amsterdam, the Netherlands, **2** Human Movement Sciences, KU Leuven: Katholieke Universiteit Leuven, Leuven, Belgium

* m.a.afschrift@vu.nl

## Abstract

Physics-based simulation generate movement patterns based on a neuro-musculoskeletal model without relying on experimental movement data, offering a powerful approach to study how neuro-musculoskeletal properties shape locomotion. Yet, simulated gait patterns and metabolic powers do not always agree with experiments, pointing to modeling errors reflecting gaps in our understanding. Here, we systematically evaluated the predictive capability of simulations based on a 3D musculoskeletal model to predict gait mechanics, muscle activity, and metabolic power across gait conditions. We simulated the effect of adding mass to body segments, variations in walking speed, inclined walking, and crouched walking. We chose tasks that are relatively straightforward to model to limit the contribution of errors in modeling the task to prediction errors. The simulations predicted stride frequency and walking kinematics with reasonable accuracy but underestimated variation in metabolic power across conditions. In particular, simulations underestimated changes in metabolic power with respect to level walking in tasks requiring substantial positive mechanical work, such as incline walking (27% underestimation). We identified two possible errors in simulated metabolic power. First, the phenomenological metabolic power model produced high maximal mechanical efficiency (average 0.58) during concentric contractions, compared to the observed 0.2–0.3 in laboratory experiments. Second, when we multiplied the mechanical work with more realistic estimates of mechanical efficiency (i.e., 0.25), simulations overestimated the metabolic power by 84%. This suggests that positive work by muscle fibers was overestimated in the simulations. This overestimation may be caused by several assumptions and errors in (the parameters of) the musculoskeletal model including its interaction with the environment or in the cost function. This study highlights the need for more accurate models of musculoskeletal mechanics, energetics, passive elastic structures, and neural control (e.g., optimality criteria) to improve the realism of human movement simulations. Validating simulations across a broad range of conditions is important to pinpoint shortcomings in model-based simulations.

**Data availability statement:** A static copy of the Simulation code and experimental data can be accessed through Zenodo: https://doi.org/10.5281/zenodo.14524619. The simulation software will be maintained in a GitHub repository https://github.com/MaartenAfschrift/PredSim_gait_conditions.The simulation software will be integrated and maintained in a GitHub repository https://github.com//KULeuvenNeuromechanics/PredSim.

**Funding:** This study was supported by FWO (Research Foundation - Flanders) postdoctoral fellowship grant 12ZP120N (salary M.A) and FWO Junior Research Grant G0B4222N (FDG). The funders had no role in study design, data collection and analysis, decision to publish, or preparation of the manuscript.

**Competing interests:** The authors have declared that no competing interests exist.

## Author summary

Our research focuses on understanding how humans walk by using computer simulations. These simulations are based on detailed models, i.e., mathematical descriptions, of skeleton, muscles, joints, and nervous system. By comparing our simulations to actual experiments where people walked under different conditions—such as carrying extra mass, walking faster or slower, or moving uphill or downhill—we evaluated how well the simulations could predict real-life movement and energy use. We found that while the walking simulations performed well in predicting the movement pattern, they underestimated metabolic energy used by the body, especially in tasks like walking uphill. Errors in simulated metabolic power likely stem from two issues. First, the metabolic energy model resulted in unrealistically high mechanical efficiency compared to experiments. Second, positive work (and as a result also net negative work) by muscle fibers was overestimated in the simulations. These findings highlight the need to improve the models so they can more accurately reflect the complexity of human movement and energy use. Ultimately, better models will help us design devices like exoskeletons and prosthetics and improve treatments for people with movement difficulties.

## Introduction

Human gait is remarkably stereotypical considering the large kinematic redundancy, i.e., the large number of possible gait patterns people can select to move forward. It is often assumed that the central nervous system exploits this redundancy to select gait patterns that are efficient and robust. However, despite decades of valuable research, it is not yet fully understood how the central nervous and musculoskeletal systems interact during walking. Physics-based simulation is a powerful approach to explore this interaction and hence to study the principles of human locomotion [1,2]. Physics-based simulations rely on a neuro-musculoskeletal model, i.e., a mathematical description of the neural and musculoskeletal systems. Such simulations allow exploring hypothetical scenarios, such as the effect of adding mass to body segments or muscle weakness on human gait [3,4]. Differences between simulated and measured gait kinematics and energetics in these scenarios are especially interesting, as such differences point to possible knowledge gaps and can lead to identification of errors in the neural control or musculoskeletal models. In this study, we simulated a broad range of walking conditions (e.g., walking with added mass, walking at various speeds, walking on a slope) with a state-of-the art musculoskeletal model and compared simulated to measured gait outcomes. We specifically chose conditions that are relatively straightforward to model, to minimize the risk that prediction errors stem from shortcomings in the musculoskeletal and neural control models, rather than inadequate modeling of the intervention itself. By contrasting simulations with experiments we aimed to identify knowledge gaps in current gait simulations.

Physics-based simulations typically assume that humans select a muscle activation pattern that result in agait pattern, from the redundant set of kinematic trajectories, that minimizes a cost function representing the cost to the human of achieving a task [1]. Such cost functions can include a variety of terms; commonly encountered terms are related to muscle fatigue (e.g., maximum muscle activations [5]), signal-dependent motor noise (e.g., activations squared [6]), musculoskeletal loading (e.g., contact forces [7]), movement smoothness (e.g., joint accelerations [8]), and metabolic power [9]. Under this assumption, simulations of human movement can be formulated as optimization problems, i.e., find muscle excitations [1,10] or (the parameters defining the) control policy [11] that minimize a cost function for a given musculoskeletal model and task constraints (e.g., moving forward at a given speed).

Metabolic power is frequently included in the cost function and is an important outcome for various applications (e.g., exoskeleton design [12]) but the accuracy of muscle energy models is still debated. Muscle mechanics [13] and energetics [14,15] are typically described by phenomenological models. Metabolic energy models phenomenologically relate Hill-type muscle states and inputs to metabolic power via heat rate (maintenance, activation and shortening/lengthening heat rate) and mechanical power, based on experiments conducted with small bundles of isolated animal muscle fibers. The mathematical models and coefficients derived from these lab experiments have been incorporated into whole-body simulations [16]. However, significant discrepancies remain between whole body metabolic power estimated using model-based simulations and measured through indirect calorimetry [17]. These discrepancies may arise from multiple modelling simplifications and errors. First, phenomenological energy models might be inaccurate for the muscles' operating conditions during walking. Converting heat rate coefficients from controlled lab experiments on single muscle fibers to whole-body human movement introduces uncertainty as the vast number of possible combinations of muscle activity, fiber length, and velocity observed during human movement are difficult to replicate in controlled lab experiments. Furthermore, it is often unclear whether these models account for both initial heat (contraction coupling efficiency) and recovery heat (phosphorylation coupling efficiency, ATP synthesis). Second, most current energy models are based on inputs and muscle states that are predicted using phenomenological Hill-type models, which are also based on measurements in controlled environments that may not capture accurately how the muscle operates during walking. In addition, muscle parameters (e.g., isometric force, tendon stiffness, ...) are typically obtained by scaling a generic model that might not well represent a specific person. Finally, rigid-body assumptions, errors in segment masses and lengths, joint definitions and interaction with the environment can all contribute to errors in simulated walking pattern and thus the underlying muscle excitations, states and energetics.

Musculoskeletal models are simplifications of the physiological system and the implications of the underlying assumptions on their ability to predict a variety of walking conditions is not well understood. Many model parameters, e.g., the weights of the different terms in the cost function, are not directly measurable. One approach to limit the effect of model errors on the simulated movement pattern, has been to add a tracking term [4,18] minimizing the difference between simulated and experimental kinematics and kinetics to the cost function. However, the reliance on experimental data confounds predictions. Another approach is to estimate parameters that cannot be measured by fitting simulations to experimental data [10,19]. This might equally well confound predictions in case of overfitting, i.e., when the data used for fine-tuning these parameters does not contain sufficient information. Hence, such approaches require extensive validation based on independent datasets. Although most studies have validated their simulations to some extent (e.g., [10,20]), a comprehensive validation across a broad range of walking conditions is still lacking.

In this study, we systematically evaluated the capability of simulations based on a muscle-driven 3D musculoskeletal model to predict experimentally observed walking kinematics, kinetics, muscle activity and metabolic power in various conditions without adapting simulation parameters. Specifically, we simulated the effect of adding mass to different body segments, variations in walking speed, walking uphill or downhill, and crouched walking and compared this to experimental data. These tasks were chosen because they are relatively simple to model and introduce variations in average muscle activity (crouched walking), net mechanical muscle work (walking uphill or downhill), and in positive and negative

mechanical muscle work (walking speed, added mass), offering insight into potential modeling errors. This systematic approach provides a benchmark for assessing the predictive accuracy of physics-based simulations. We used a previously published trajectory optimization workflow and 3D musculoskeletal model with 31 degrees of freedom driven by 92 Hill-type actuators [21]. Gait patterns are calculated by minimizing a multi-objective cost function. Cost function weights were hand-tuned to obtain physiologically plausible walking kinematics, kinetics, and muscle activations at 1.3 m/s [10]. Here, we performed a comprehensive evaluation of this simulation approach. We think that results of our evaluation are of broader interest as many modeling choices (e.g., phenomenological Hill-type muscle models, assumption of optimality) are common in physics-based simulation of human motion.

## Methods

In order to identify knowledge gaps in human gait simulations, we systematically compared the simulated and measured effect of simple mechanical interventions on gait kinematics, kinetics and energetics.

### Musculoskeletal model

Simulations were based on the 3D musculoskeletal model presented in [21]. In short, the model is based on an Open-Sim musculoskeletal model with 31 degrees of freedom (dof; pelvis as floating base 6 dof, 3 dof hips, 1 dof knees, 2 dof ankles, 1 dof toes, 3 dof lumbar joint, 3 dof shoulders and 1 dof elbows) and 92 muscles actuating the lower limb and lumbar joints, eight ideal torque motors actuating the shoulder and elbow joints, and six contact spheres per foot [10,22]. We added passive exponential stiffness and linear damping to the lower limb and lumbar joints to model ligaments and other soft tissues spanning the joint. Muscle-tendon paths and moment arms of the OpenSim model were approximated by polynomial functions (muscle-tendons length and moment arms as a function of joint angles) to improve computational speed [23]. We used Raasch's model to describe muscle excitation-activation coupling [24] and a Hill-type muscle model to describe muscle-tendon interaction and the dependence of muscle force on fiber length and velocity [25]. We modeled skeletal motion with Newtonian rigid-body dynamics and we used a smooth approximation of the Hunt-Crossley foot-ground contact model [26]. The relation between muscle states and metabolic power was modeled as in Bhargava et al. [14], as this energy model had the highest correlation with experimental data in a previous simulation study [17]. More details about the musculoskeletal model can be found in a previous publication [21].

### Optimal control simulations

We predicted human movement based on the assumption that humans select a muscle excitation pattern that minimizes a cost function. As a result, we formulated simulations as optimal control problems. We identified muscle excitations and the gait cycle duration that minimized a cost function subject to constraints describing muscle and skeleton dynamics, that limbs cannot cross each other and task constraints, i.e., walking speed, left-right symmetry, and additional constraints for some of the gait conditions (see below). Similar as in previous publications [10,21], our cost function consisted of the time-integral of a weighted sum of squared metabolic power ($\dot{E}$), muscle activations ($a$), joint accelerations ($\ddot{q}$), passive torques ($T_p$), and excitations of the ideal torque motors at the arm joints ($e_a$):

$$J = \frac{1}{d}\int_0^{t_f} \left( \underbrace{w_1 \parallel \dot{E} \parallel_2^2}_{\text{Metabolic power}} + \underbrace{w_2 \parallel a \parallel_2^2}_{\text{Muscle activity}} + \underbrace{w_3 \parallel \ddot{q} \parallel_2^2}_{\text{Joint acc}} + \underbrace{w_4 \parallel T_p \parallel_2^2}_{\text{Passive torques}} + \underbrace{w_5 \parallel e_{\text{arms}} \parallel_2^2}_{\text{Arm excitations}} + \underbrace{w_6 \parallel u_{\text{slack}} \parallel_2^2}_{\text{Slack controls}} \right) \mathrm{d}t, \tag{1}$$

To avoid singular arcs [27], we added penalty for slack controls (sum of squared time derivatives of normalized muscle forces and muscle activations) with a small weight to the cost function. We used the weights ($w_1$- $w_6$) as published in [21] (see Table A in S1 File). More details about the optimal control problem formulation can be found in previous work

[21]. We performed simulations in our PredSim framework that relies on OpenSim to derive skeleton dynamics and uses MATLAB and CasADi [28] to formulate the optimal control problem [29]. We applied direct collocation using a third order Radau quadrature collocation scheme, used algorithmic differentiation to compute derivatives, and solved the resulting nonlinear programming problem with IPOPT [30].

## Modeling experimental walking conditions

In short, we ran simulations in an attempt to replicate a broad range of experimental studies on human locomotion that studied the effect of added mass [31–33], walking uphill or downhill [17,34–37], variations in imposed walking speed ([35,38], Table 1) and crouched walking [34] (i.e., walking under a virtual ceiling). We simulated the effect of added mass at the feet, ankles, shanks, knees, thighs, pelvis and trunk on human walking. We modeled the added mass as a point mass that is rigidly attached to the specific segment by adapting the segment's mass, center of mass location, and inertia. We simulated walking on a slope by rotating the direction of gravity in the simulation environment. Walking speeds were imposed by constraining the average forward speed of the pelvis. We also simulated walking on slopes and

**Table 1. Description of all studies we replicated in simulation.**

| Study | Partici-pants | Slope | Added mass [kg] | Walking speed | Measurement metabolic power | Mean body mass [kg] | Mean height [m] | Mean age [years] | Method data extraction |
|---|---|---|---|---|---|---|---|---|---|
| Gome-nuka et al. | 10 | 0%<br>7%<br>15% | 25% body mass, backpack | 2, 3, 4, 5, 6 km/h | O2 and CO2 metabolic rate standing subtracted | 71.6 | 1.78 | 23 | Digitized figures |
| Browning et al. | 5 | 0% | foot (4, 8)<br>tibia (4, 8)<br>femur (8, 16)<br>pelvis (4, 8, 16 kg) | 1.25 m/s | O2 and CO2 metabolic rate standing subtracted | 74.16 | 1.82 | | Digitized figures |
| Schertzer et al. | 8 | 0% | ankle (1, 2, 4)<br>knee (1, 2, 4)<br>back (2, 7.1, 10.1, 16.1, 22.1) | 4-5-6 km/h | based on gas exchange but details not described | 74.88 | 1.78 | 27 | Digitized figures |
| Huang et al. | 8 | 0% | 6.8 kg - 20.4 kg backpack | 1.25 m/s | O2 and CO2 metabolic rate standing subtracted | 71.1 | [0.99 leg length] | 19-26 years | Based on regression equation |
| Koelewijn et al. | 12 | level, 8% and -8% | | 0.8 and 1.25 m/s | O2 and CO2 metabolic rate standing subtracted | 70 | 1.73 | 24 | Based on raw data processed with addbiomechanics |
| Van Der Zee et al. | 10 | level | | 0.7 - 0.9 - 1.1 - 1.25 - 1.4 - 1.6 -1.8 - 2 m/s | no | 73.5 | 1.76 | 24 | Based on raw data pub-lished in addbiomechanics dataset |
| McDonald et al. | 10 | level, 6%, 12%, 18% and 24% | | 1 m/s | O2 and CO2 metabolic rate standing subtracted | 69.6 | 1.70 | 31 | Based on raw data |
| Abe et al. | 11 | level, -5% and 5% | | 0.67, 0.86, 1.06, 1.25,1.44, 1.64, 1.83, 2.03, 2.42, 2.62, 2.81, 3 m/s | VO2 measurement, Standing VO2 Computed CO2 rate and computed metabolic energy | 58.9 | 1.70 | 20 | Digitized from figures |
| Strutzen-berger et al. | 15 | -12, -6, 0, 6, 12% | | 1.1 m/s | | 73.1 | 1.77 | 25 | Based on Table 1 |

crouched walking to replicate an experimental study that provided insight in the relative importance of minimizing metabolic power versus minimizing muscle activity [34]. In that experiment, crouched walking was achieved by instructing the participants to walk with an upright trunk and avoid contact with a virtual ceiling at 93% body height. We implemented this in our simulations by constraining the maximal pelvis vertical coordinate to be below 93% of the maximal value in the unconstrained simulation. In our simulations, we did not impose a specific form of locomotion, allowing the model to freely determine the gait pattern (walking, grounded running, running). This contrasts with the experiments by Abe et al., where participants were instructed to walk at treadmill speeds of 2.02 m/s and below and to run at speeds of 2.4 m/s and above [35]. Similarly, in the study by Van der Zee et al., participants were constrained to walking at speeds up to a maximum of 2 m/s [38].

### Normalization of data

To facilitate comparison between studies, experimental data were converted to dimensionless units. We used body mass (m), leg length (l), and gravitational acceleration g = 9.81 m/s$^2$ for normalization. Step frequency was normalized by $\sqrt{g/l}$, joint moments by $mgl$, ground reaction forces by $mg$, and (metabolic) power by $mg^{1.5}l^{0.5}$. Note that in most studies, individual data were not provided and therefore we used the average body mass and height for normalization. We then computed the corresponding non-normalized values for a model with a body mass of 62 kg and stature of 1.70m – the mass and length of our model - to facilitate interpretation of the results.

### Comparison of simulations and experimental observations

To evaluate the predictive capability of the simulations, we compared simulated and measured stride frequencies, joint angles and moments, ground reaction forces, and metabolic powers. For studies that published raw data, we used addBiomechanics [39] to extract joint angles and moments. When raw data was unavailable in the experiments, we digitized stride frequency and metabolic power from published figures.

Our evaluation is largely based on relative changes in metabolic rate when comparing walking conditions (i.e., change in metabolic power with respect to a reference condition) because differences in how different studies measured resting metabolic rate hinder the comparison of absolute values of metabolic power. For most comparisons, we chose as the reference condition level walking at 1.1 m/s without added mass and without any constraints on stride frequency and reported the change in metabolic power with respect to this condition. If level walking at 1.1 m/s was not included in the experiment, we selected the condition closest to this walking speed as the reference. To evaluate the effect of added mass on the metabolic power, we chose a different reference, i.e., level walking at the same speed without added mass. The reader should therefore carefully interpret metabolic power results. Absolute values of measured and simulated metabolic powers can be found in the appendix (Fig H in S1 File).

We simulated walking on slopes and crouched walking in order to replicate an experimental study that provided insight in the relative importance of minimizing metabolic power versus minimizing muscle activity [34]. We evaluated if the simulation model could predict the changes in metabolic power and average activity of a subset of muscles (in the experiment based on electromyography) when increasing the slope or when walking in crouch. Additionally, based on the value of the cost function in the simulation of uphill and crouch walking, we evaluated whether the simulation model would also prefer tasks that avoided overburdening muscles (crouch walking) at the expense of higher metabolic power (i.e., lower cost function value indicates that the task is preferred by the simulation model).

We quantified the relation between simulated and measured changes in stride frequency and gait cycle average metabolic power (referred to as metabolic power) across different walking conditions by computing the explained variance ($R^2$), root-mean square error (RMSE), and slope of the relation using a linear least squares fit. The slope was used to identify over or underestimation of metabolic power. For trajectory outcomes (joint kinematics and kinetics) we discussed the differences between experiments and simulations qualitatively.

## Mechanical efficiency of the metabolic energy model

The efficiency of net mechanical work done in the gait simulations was computed as the predicted net mechanical work by muscle fibers divided by the predicted net metabolic energy in one gait cycle. The net metabolic energy is the total metabolic energy minus the basal metabolic energy. The efficiency of positive mechanical work was computed as the predicted positive mechanical work done by muscles fibers divided by the predicted net metabolic energy. Mechanical work done by fibers is the time integral of muscle fiber power over a full stride. Muscle fiber power was computed as the product of the contraction velocity of the muscle fiber and its force. As the efficiency of net and positive mechanical work was unrealistically high compared to experiments [16] we also computed the maximal mechanical efficiency of muscle fibers in our model. We did this by computing muscle fiber power and metabolic power in isokinetic contractions at multiple fiber contraction velocities while the muscle fiber was maximally activated and when the muscle fiber was at the optimal length. In addition, due to the unrealistic mechanical efficiency obtained with the Bhargava metabolic energy model, we also evaluated simulations using a model that only accounts for the energy cost of mechanical power and assumes concentric power has an efficiency of 0.25 and eccentric power has an efficiency of -1.2 (Margaria model [40]). We used this energy model to analyze the metabolic power in the default simulations [14] (i.e., post processing the simulation results with cost function (eq. 1). In addition, we also ran new simulations with metabolic power computed with this energy model in the cost function instead of the Bhargava model [14].

## Results

### Simulations capture differences in stride frequency between walking conditions

Our simulations captured the experimentally observed effect of most walking conditions on stride frequency (Fig 1). The simulations captured changes in stride frequency for variations in walking speed ($R^2 = 0.99$, 10% over estimation, rmse = 0.03 Hz) and variations in added mass ($R^2 = 0.99\%$, 30% underestimation, rmse = 0.03 Hz). The simulations predicted that walking on a slope had a negligible effect on stride frequency ($R^2 = 0.03$, underestimation 90%, rmse = 0.04 Hz). Moreover, the simulation not only predicted the changes in stride frequency in the different gait conditions but also captured the absolute values (Fig A in S1 File).

### Simulation capture some, but not all, effects of walking conditions on joint kinematics and kinetics

We evaluated if the simulation could adequately predict measured joint kinematics and kinetics for walking at various speeds [38] and walking on a slope [17]. The predicted joint kinematics and kinetics are reasonably accurate for walking

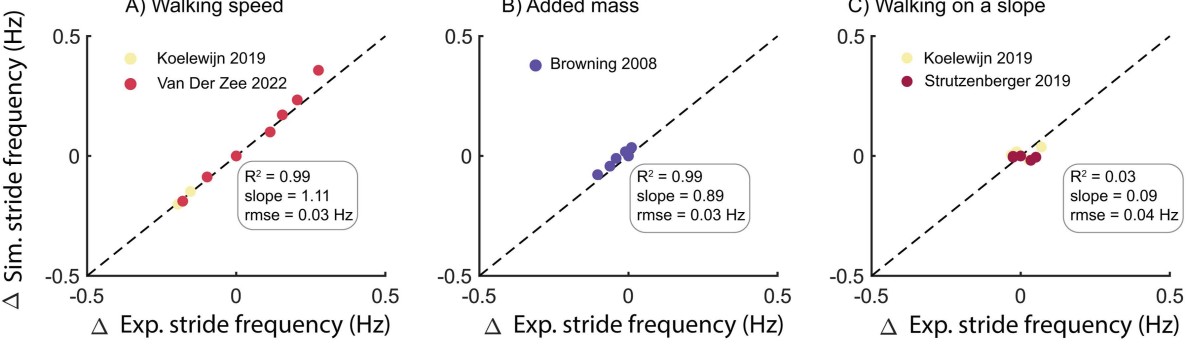

**Fig 1. Predicted changes in stride frequency from level walking at 1.1-1.2 m/s with changes in walking speed (A), added mass at various locations on the body (B), and walking on a slope (C).** Experimental data were converted to non-dimensionless units and subsequently converted for the anthropometry of the simulation model.

at 1.3 m/s (the condition that was used to fine-tune the weights in the cost function [10]) except for the ankle joint kinematics during stance which underestimated the dorsiflexion during midstance and plantarflexion at push-off (Fig B in S1 File). As we were mainly interested in the experimental conditions that were not used to fine-tune the weights in the cost function, we focused on evaluating whether simulations capture alterations with respect to walking at self-selected speed.

For walking at various speeds, the simulations captured changes in kinematics for speeds between 0.7 and 1.6 m/s. The simulation adequately predicted the increase in ankle plantarflexion angle at push-off and knee flexion at initial stance with increased walking speed (Fig B in S1 File). However, the simulation overestimated changes in ankle and knee kinematics when gait speed was further increased to 2 m/s. In contrast with the experiments, the simulations predicted a more crouched position at push off at higher gait speeds (Fig B in S1 File). The observed increase in hip flexion moment and extension moment with faster walking was predicted accurately (Fig C in S1 File). The simulations failed to capture the increase in plantarflexion moment at push-off with increasing gait speed (Fig C in S1 File). Instead, they predicted large changes in knee moments at the highest gait speeds that were not observed experimentally. The simulations also failed to capture the increase in magnitude of the anterior-posterior component of the ground reaction force with increased gait speed (Fig D in s1 File). The discrepancies between experiments and simulations may partly stem from the transition to a (grounded) running gait in simulations at speeds of 1.8 m/s and above, in contrast to the imposed walking gait at all evaluated speeds (up to 2 m/s) in the experiment by Van der Zee et al. [38]. Additional details on the walk-to-running transition are provided in Fig O in S1 File.

For walking on a slope, the model captured the changes in ankle, knee and hip kinematics reasonably well for walking on a positive or negative slope (Fig E in S1 File). Similar to experimental data, the simulations showed increased plantarflexion angle at stance-swing transition in uphill walking, increased knee flexion at initial stance and more hip flexion in general. The model also captured most changes in joint kinetics for walking on a slope (Fig F in S1 File) with increased hip flexion moment for downhill walking and increased hip extension moment for uphill walking. The model failed to capture the increase in knee flexion moment during early stance in downhill walking.

## Simulations underestimate changes in metabolic power due to changes in walking conditions

We evaluated the simulations' ability to predict changes in metabolic power across various conditions. In general, simulations underestimated the changes in metabolic power compared to walking at 1.1 m/s ($R^2 = 0.91$, 15% underestimation, rmse = 57W). The simulations predicted the effect of varying gait speed on metabolic power with reasonable accuracy ($R^2 = 0.92$, 10% underestimation, rmse = 53.9 W). The simulations particularly underestimated the increase in metabolic power at high gait speeds (Fig 2C). The simulations underestimated the increase in metabolic power due to added mass ($R^2 = 0.81$, 21% underestimation, rmse = 17.3 W). The model underestimated the effect of slope on metabolic power both for positive and negative slopes ($R^2 = 0.93$, 27% underestimation, rmse = 59 W).

We also compared the absolute values of the simulated and measured metabolic powers across conditions (Fig H in S1 File). Experimentally reported measures of metabolic power for a given walking condition (e.g., level walking at 1.1 m/s) varied between experimental studies, indicating that differences in measurement equipment or protocol (e.g., subtraction of basal metabolic power) confound this analysis. This complicated the interpretation of the simulations results and might explain why we found, depending on the experimental study, an over- or underestimation of the simulated metabolic power for a given gait condition.

## Simulations capture the activation, but not the energetic cost of walking

We simulated walking on a slope and in crouch to determine whether our simulations capture the experimentally observed trade-off between increased muscle activity (walking in crouch) and metabolic power (walking on a steep slope) [34]. We first compared the simulated and measured changes in metabolic power and activation cost (i.e., the average activity of a subset of muscles reported by [34]) for crouch walking and walking on a slope. We found that the

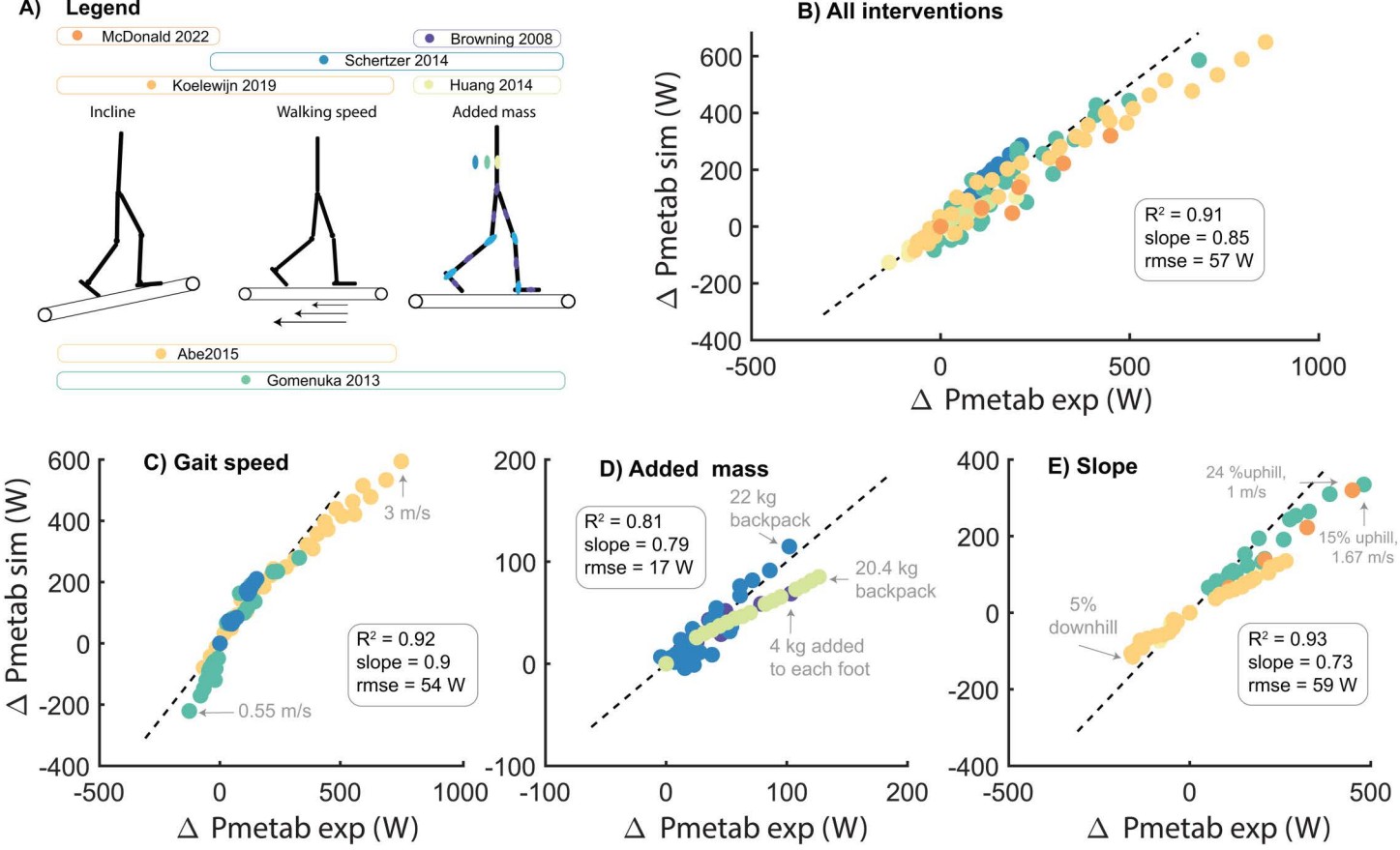

**Fig 2. Predicted changes in gait cycle average metabolic power (Pmetab) from level walking at 1.1-1.2 m/s with changes in walking speed (C), mass added to different body segments (D) and walking on a slope (E).** The colors represent the subject-mean data reported in the different studies (A). Experimental data were converted to non-dimensionless units and subsequently converted to the anthropometry of the simulation model.

simulations substantially underestimated the increase in metabolic power with increased slope and crouch (Fig 3B). However, unlike the metabolic power predictions, the activation cost was simulated with reasonable accuracy for both walking conditions (Fig 3C).

Next, we examined the cost function value across gait conditions to assess whether the simulations captured the preference of participants for crouch over slope walking for a slope gradient above 12%-18%. We found that the cost function value of crouch walking was lower than for slope walking across all gradients tested (6%, 12%, 18%, 24%) (Fig 3D). The metabolic power component of the cost function was lower for crouched walking than for slope walking at all tested slopes above 0% whereas the experimentally measured metabolic power of crouch walking is similar to walking with a slope of 12% incline (Fig 3E). Whereas the activation cost of the subset of muscles that were experimentally assesses agreed well between simulations and experiments (Fig 3C), the muscle activation component in the cost function was lower for crouch walking than for all tested slopes (Fig 3F). The activation cost of a subset of muscles might thus not be representative for all muscles. The muscle activation component in our cost function was not dependent on muscle volume (i.e., the activation of muscles with small and large volumes contributed equally in the cost function). When we postprocessed the results using relative muscle volumes from [34], we found that this weighting altered the activation component in the cost function for both crouch and slope walking (Fig 3G).

**Fig 3. Predicted changes in gait cycle average metabolic power (Pmetab) and activation cost (ActCost) for crouched walking (red) and walking on a slope (blue).** The simulations underestimated the increase in average metabolic power for slope and crouch walking (B) and estimated the increase in activation cost for slope and crouch walking with reasonable accuracy (C). The cost function in the simulation could not capture the observed transition from slope to crouch walking (D). Similar as in experiments, the cost function term related to metabolic power (integrated metabolic power squared multiplied by weight) was smaller for crouch walking than for walking on a slope (E). It was harder to compare the activation related component of the cost function (integrated muscle activations squared multiplied by weight) between experiments and simulations as only a subset of muscles was used in the experiment to compute activation related costs and all muscles were used in the simulation. We found that the activation related cost in simulations was not higher for crouch walking compared to walking on a slope (F). This was slightly different when computing the volume-weighted muscle activations (G).

## Unrealistically high mechanical efficiency of muscle contraction

We evaluated the mechanical efficiency of the muscle model to better understand the underestimation of changes in metabolic power across all walking conditions. The simulated ratio of positive fiber work to metabolic energy was close to 0.5 for some conditions of fast and uphill walking (Fig 4A). The simulated ratio of net muscle work to metabolic energy was up to 0.38 for some conditions of uphill walking and walking with added mass (Fig 4B).

To investigate the maximal mechanical efficiency - for positive and negative work, we conducted additional single-muscle simulations where simulated metabolic power and mechanical work was evaluated for muscle contractions at optimal fiber length with maximal activation and with varying contraction velocities (Fig 4C–4F). The maximum mechanical efficiency for positive mechanical work ranged from 0.47 to 0.64, with an average maximal efficiency of 0.58 across all muscles (Fig 4C, 4D). The minimal mechanical efficiency of negative mechanical work, which reflects the energetic cost of dissipating mechanical energy, ranged from $-2 \cdot 10^4$ to $-0.5 \cdot 10^4$ (Fig 4E, 4F).

## Metabolic energy models are too efficient in performing mechanical work

Given the high mechanical efficiency observed in our muscle energy model, we explored if an alternative metabolic energy model with mechanical efficiency based on in vivo experiments can improve the prediction of metabolic power (Margaria model) [40]. This model assumes that muscle fibers have an efficiency of 0.25 when performing positive

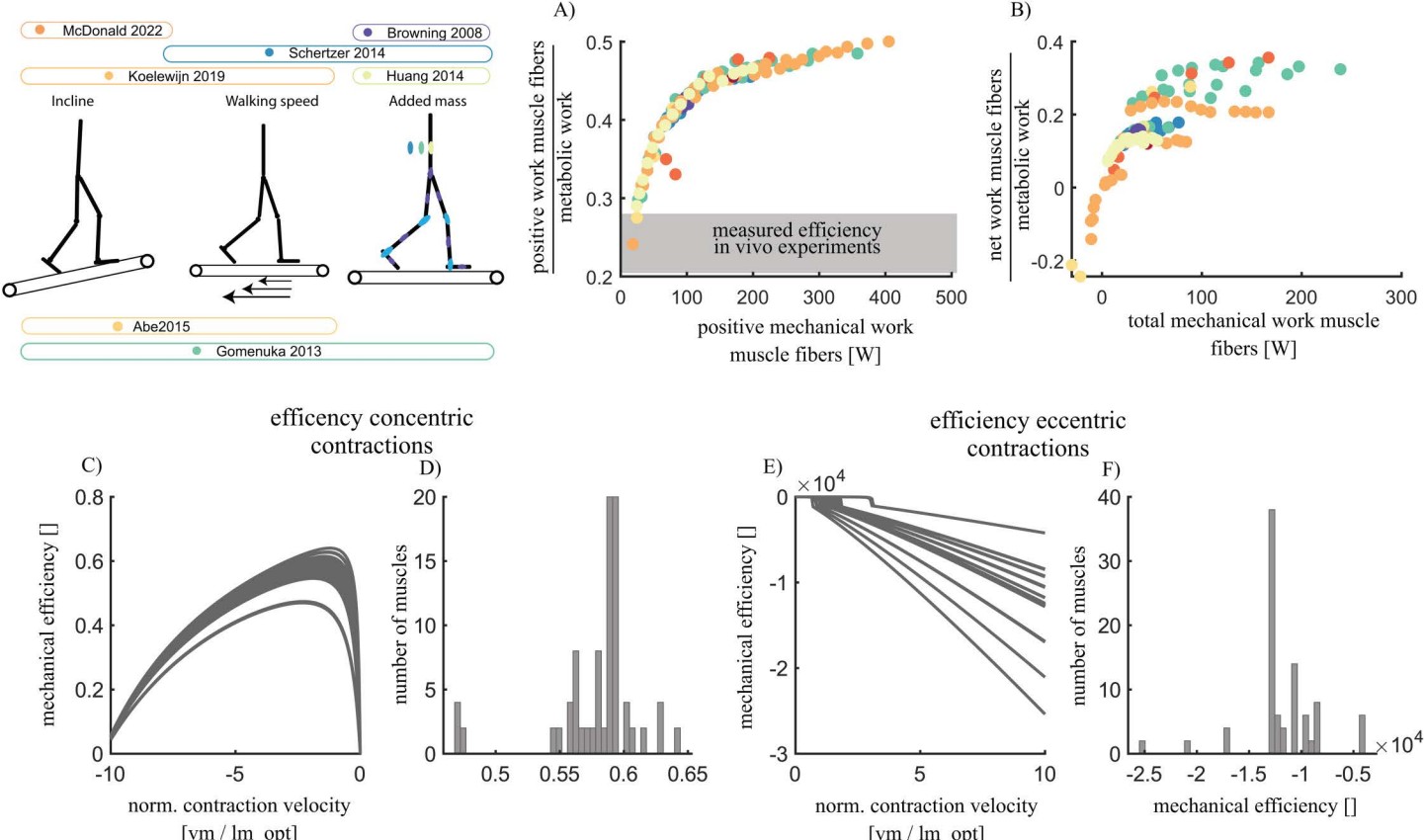

**Fig 4. The Bhargava metabolic energy model has an unrealistically high maximal mechanical efficiency.** The ratio of total simulated positive muscle fiber work and simulated metabolic energy was up to +/- 0.5 in the simulated walking conditions **(A)** and the ratio of simulated net muscle work and metabolic energy was up to +/- 0.35 **(B)**. We performed an analysis of the maximal mechanical efficiency of the 92 muscles in our model by computing muscle fiber power and metabolic power in isokinetic contractions at different fiber contraction velocities while the muscle fiber was maximally activated and operated at its optimal length. In this analysis we found mechanical efficiencies of up to 0.6 for concentric contractions with a mean of 0.58 for the 92 muscles **(C-D)** and -20000 for eccentric contractions with a mean of -14000 for the 92 muscles in our model **(E-F)**.

mechanical power (concentric contraction) and -1.2 efficiency when performing negative mechanical power (eccentric contraction) [40]. First, we post-processed the simulations with the Margaria model to evaluate if it resulted in more realistic metabolic powers. Second, we predicted new walking motions with a cost function based on the Margaria instead of the Bhargava energy model to evaluate whether this resulted in more realistic walking motions and simulated metabolic powers.

When post-processing the simulations with the Margaria energy model, we found that it overestimated the increase in metabolic power with changes in walking speed ($R^2 = 0.9$, 100% overestimation, rmse = 266 W, Fig 5B). However, we found reasonably accurate estimates of the increase in metabolic power due to added mass ($R^2 = 0.83$, 23% overestimation, rmse = 22 W) and walking on a slope ($R^2 = 0.95$, 48% overestimation, rmse = 103W, Fig 5C and 5D).

When we incorporated the Margaria energy model, instead of the Bhargava energy model, into our multi-objective optimization, the model accurately predicted changes in metabolic power with varying walking speeds (Fig 6A, 6B). However, the simulations no longer adequately captured experimentally observed stride frequency and joint moments (Fig 6C–6I).

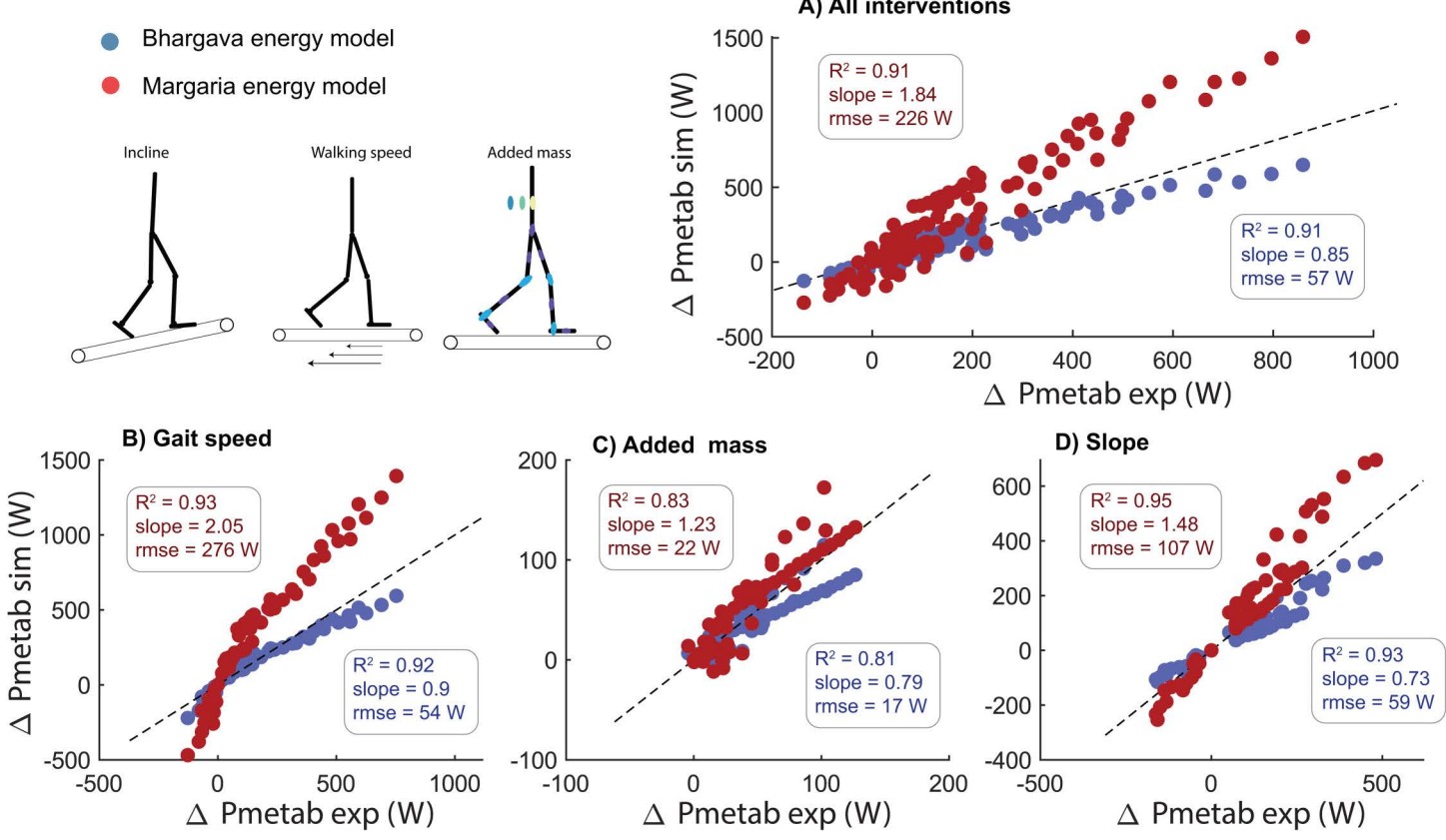

**Fig 5. Postprocessing the simulations with the Bhargava energy model (blue) or Margaria energy model that only accounts for the metabolic power of mechanical muscle fiber power (red) has a large influence on the predicted change in metabolic power.** The Bhargava energy model systematically underestimated the change in metabolic power, while the Margaria energy model systemically overestimated the change in metabolic power with variations in walking conditions.

## Discussion

This study evaluated the capability of physics-based simulations of musculoskeletal models to predict human gait across various conditions, including variations in speed, slope and added mass, to gain insight in limitations in our current understanding of the mechanics and energetics of human walking. While the simulations reasonably predicted changes in spatio-temporal parameters, joint kinematics and kinetics under varying conditions (Fig 1), they underestimated changes in gait cycle average metabolic power (Fig 2). Errors in metabolic power predictions are not simply due to errors in the metabolic energy model since the simulated muscle mechanical powers would yield unrealistically high metabolic power estimates when assuming realistic muscle efficiencies (Fig 5). This demonstrates that while physics-based simulations can reasonably predict variations in human walking joint kinematics and joint moments, substantial limitations exist in the prediction of metabolic power.

The underestimation of changes in metabolic power with respect to level walking at 1.1 m/s, particularly in conditions with added mass and for incline walking (Fig 2), appears related to the unrealistic mechanical efficiency of muscles in the phenomenological energetics models (Fig 4). The Bhargava energy model yields muscle mechanical efficiencies between 0.47 and 0.64 averaging 0.58. While these values align with previous modeling studies [15,41] and are similar to other energy models that model heat rate (Fig N in S1 File, [42]), they are substantially higher than the maximal

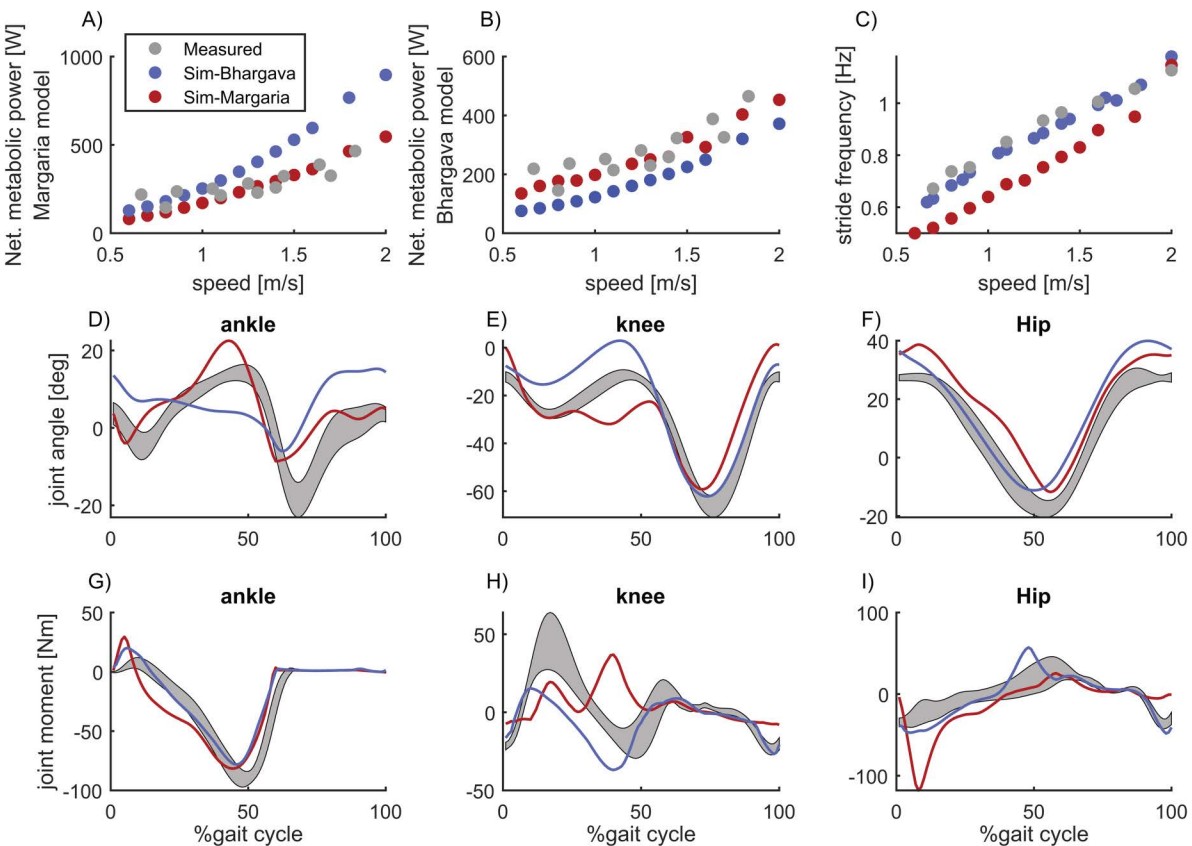

**Fig 6. Simulated gait with the Bhargava energy model (blue) or the Margaria energy model (red) in the (multi objective) cost function.** The simulations with the Bhargava energy model in the cost function overestimated the effect of walking speed on metabolic power computed with the Margaria energy model but not metabolic power computed with the Bhargava energy model. While the simulations with the Margaria energy model in the cost function resulted in realistic metabolic powers for both energy models, the walking simulations deviated strongly from measured kinematics and kinetics (D-I, data walking at 1.3 m/s).

0.2-0.3 efficiency observed in laboratory experiments on small bundles of muscle fibers [16] and task-average efficiencies in whole-body exercises predominantly involving positive muscle fiber work such as cycling [43,44]. This discrepancy explains the underestimation of the metabolic cost for uphill walking in the simulations, where net positive mechanical work due to gravity is substantial (Fig 2 and Fig O in S1 File). The model's high mechanical efficiency might result from errors in converting heat rate coefficients derived from in vitro experiments to simulations. Moreover, the high mechanical efficiency might also stem from only modelling initial heat and ignoring the energy cost of recovery heat (i.e., ATP resynthesized via oxidative pathways, with a phosphorylating coupling efficiency of 0.5-0.6 [16,45]). Assessing metabolic power during single-joint tasks with variable mechanical power and contraction forces could provide data for estimating heat rate coefficients for in vivo human muscle function [46].

While muscle mechanical efficiency in humans has never been reported to exceed 0.25, this value is mainly based on experiments with isolated muscle fibers [16] or derived from whole-body exercises with supposedly predominantly positive work [47]. This raises the question whether 0.25 efficiency is also applicable for muscle contractions in our simulations. Isolated muscle fiber experiments are typically done in very specific conditions (i.e., specific operating length and velocity of muscle fibers, stimulation patterns, temperatures) with tetanic activation. The stimulation amplitude and frequency in most muscle fiber experiments are different from those generated by the nervous system, potentially leading to an

underestimation of maximal efficiency [45]. The accuracy of mechanical efficiency estimates derived from whole-body movements with predominantly positive mechanical work (e.g., cycling) can also be questioned. The mechanical efficiency of 0.22-0.23 in cycling, computed as the net mechanical work at the crank divided by the (net) metabolic energy, does only account for the net mechanical work done by muscles fibers. This implicitly assumes that during cycling no negative muscle work is performed. Recent simulations show that this assumption may very well not be correct, and that actual mechanical efficiency is substantially higher than the efficiency measured (~0.28 vs 0.17, respectively, pre-print: [48]). Furthermore, mechanical work for other tasks, for example by the heart and respiratory muscles to increase blood and air circulation, is not accounted for in these analyses. In addition, the estimates based on cycling experiments should be interpreted as the task-average rather than maximal efficiency of muscles as it is unlikely that muscles perform work at the energetically most efficient length and velocity during the full cycling motion. Not accounting for all mechanical work has most likely a small influence on the estimated mechanical efficiency (i.e., in the order of a few percent) as the mechanical power for increased blood and air circulation when cycling is in the order of 10W and average mechanical power at the crank is in the order of 200W. Hence, the maximal mechanical efficiency of concentric contractions might be slightly higher than 0.25, but the reported 0.58 maximal efficiency in our model and 0.4 efficiency in uphill walking simulations remains implausibly high.

The inaccuracies in predicting metabolic power might not only stem from errors in the energy model, but also from inaccuracies in modelling the musculoskeletal dynamics. Metabolic power computed with the Margaria model is expected to underestimate measured values as it only accounts for mechanical efficiencies of concentric and eccentric contractions and does not account for the cost of isometric contractions. Nevertheless, the Margaria model overestimated the metabolic power in the simulated gait patterns (Fig 5). This indicates that positive work (and as a result also negative work) by muscle fibers was overestimated in the simulations, particularly at higher walking speeds. For instance, in our simulation of level walking at 1.3 m/s, positive fiber work was 101 J/stride and negative fiber work was -56 J/stride (details in Fig M in S1 File), resulting in an estimated cost of transport (COT) of 5.02 J/kg/m, which is substantially higher than the experimentally measured 3.0-3.5 J/kg/m [17,32,35]. The overestimation of mechanical energy is not specific for the simulation approach used in this study but has also been reported in previous forward [41,49] and inverse simulations [17]. The implausibly high positive and negative muscle fiber work likely arises from a combination of assumptions and inaccuracies in the musculoskeletal model. In the following sections, we discuss some contributors that we consider likely, although this list is certainly not exhaustive.

Excessive energy dissipation in ground contact and rotational dampers might contribute to the over-estimation of negative (and hence positive) muscle fiber work. Energy dissipation in the ground contact model was higher in simulation compared to estimates from experiments (30 J/stride in simulation versus 7.6 J/stride in experiments [50]). Similarly, energy dissipation in rotational dampers in the simulation was also higher compared to estimated soft tissue energy dissipation in experiments (17 J/stride in simulation versus ~10 J/stride in experiments [51]). A less compliant ground contact model [52] reduced dissipation in the ground contact to 15 J/stride but led to a higher estimated COT of 6.46 J/kg/m with the Margaria energy model for level walking at 1.3 m/s due to increases in positive and negative fiber work.

The high net positive mechanical work done by muscle fibers in simulation might also be due to errors in modeling muscle-tendon mechanics. Accurately modelling the interaction between the tendon in series with the muscle fiber is essential for estimating muscle fiber work [53]. While our model includes compliant tendons, errors in modelling tendon stiffness might lead to an underestimation of energy stored in tendons and an overestimation of muscle work. Recent studies indicate that simulations with a more compliant Achilles tendon produce ankle kinematics more consistent with experimental data [52], suggesting that the Achilles tendon in our model may be too stiff. While ultrasound studies might provide some insight in the errors in modelling the storage and release of energy in tendons, it is challenging to accurately measure tendon and fiber kinematics and forces during human movement. Similar as with tendon compliance, inaccurate modelling of the energy storage and release in the muscles' connective tissue (i.e., parallel elastic element) might result in overestimation of mechanical work done by muscle fibers [54]. Additionally, certain mechanisms, such as titin's elasticity and residual force enhancement following eccentric contraction [55], are not modeled and may influence muscle fiber

mechanical work. Finally, we simplified the complex 3D anatomy of a muscle to a massless 1D actuator with variable pennation angle and constant width. While neglecting muscle mass might influence the simulated mechanical work, incorporating muscle mass would likely result in even higher estimates of positive mechanical work.

The optimality assumption and the particular cost function used might also contribute to errors in gait mechanics and metabolic power. Both the terms and their relative weights in the cost function shape the predicted gait pattern and thus the predicted metabolic power [1]. It is unlikely that the objective function used here is an accurate representation of the gait-related cost minimized by humans for two reasons. First, the terms and weights in the cost function were determined based on a trial-and-error approach based on the realism of the simulated walking pattern at 1.3 m/s. It is unlikely that experimental data of walking at 1.3 m/s contains enough information to identify the cost function resulting in a considerable risk of overfitting. Instead, the cost function should be identified from a comprehensive dataset spanning a range of gait conditions that allow dissociating the different cost function terms. We found that the relative contribution of different terms (e.g., muscle activations vs. metabolic energy) varied across our simulated gait tasks (Fig P in S1 File). Identification of the cost function is further complicated by the correlations between different candidate cost function terms. Yet, progress is to be expected from using data of carefully designed experiments [34]. Second, identification of the cost function is complicated by modeling errors. The specific cost function terms and weights might be required to obtain realistic gait patterns in the presence of errors in the musculoskeletal or energy models. For example, there is no experimental evidence for squaring the metabolic rate or adding joint accelerations. Maybe such terms might not be needed with better metabolic cost models or when accounting for uncertainty (e.g., due to sensorimotor noise). It is recommended to replace activation terms by volume-scaled activation terms as the activation cost is sensitive to splitting muscles in different bundles whereas volume-scaled activation is not [56]. Finally, inaccuracies in the cost function, or in the neural control model in general, may also affect how co-contraction is modeled, which in turn influences simulated mechanical work [41]. Increased co-contraction increases energy dissipation in antagonistic muscles, which has to be compensated by increased positive work in agonistic muscles. It is however unlikely that high levels of co-contraction cause the large amount of positive and negative mechanical work in our simulation as co-contraction was implicitly minimized in our cost function (activation squared and metabolic energy term).

It seems unlikely that changing the cost function could yield both realistic walking kinematics and accurate predictions of positive and negative muscle fiber work in our simulations. Minimizing metabolic energy with the Margaria energy model (based on efficiencies of positive and negative fiber work) produced realistic metabolic costs for walking at different speeds (Fig 6A, red and gray dots), but at the expense of poor kinematic and kinetic predictions (Fig 6C–6I). The Margaria model likely underestimates the metabolic energy due to not accounting for the cost of isometric contractions, and therefore our simulations based on minimizing metabolic energy according to Margaria indicate that errors in musculoskeletal dynamics rather than only the cost function cause unrealistically high muscle work. Similar trends are evident in other simulation studies that applied the Margaria model to estimate the metabolic cost of walking based on experimental data of walking kinematics. For example, Koelewijn et al. [17] also reported overestimations of metabolic power during level walking, consistent with an overestimation of positive and negative fiber work. Moreover, they also found that the Bhargava [14] and Umberger [15,42] models yielded realistic metabolic power during level walking, but substantially underestimated metabolic power for slope walking. Given that Koelewijn et al. used an inverse approach with a different cost function to solve the muscle redundancy problem, similarities between their and our findings further confirm that the apparent agreement between simulated and measured metabolic power during level walking (Fig 2) likely stems from overly high mechanical efficiency of the metabolic energy model masking the unrealistically high predicted muscle fiber work.

Although we selected tasks that were relatively easy to model, our simulations did not fully capture the experimental conditions. First, we used treadmill walking data to benchmark the simulations at various gait speeds but did not explicitly model walking on a treadmill. The treadmill belt speed is often not perfectly constant (e.g., belt speed is often lower than desired at heel strike) due to interactions between the human and treadmill and accounting for these interactions

would require us to model the treadmill as well. In addition, walking on a treadmill alters sensory inputs (e.g., from vision) but our simulation framework does not capture sensorimotor control. Second, in the studies of Van der Zee et al. [38] and Abe et al. [35], participants were required to walk at all speeds up to 2 m/s and to run at higher speeds, whereas the simulations were free to adopt either gait pattern at each speed (Fig O in S1 File). This discrepancy mainly affects comparisons around the walk-to-run transition (~2 m/s). Third, experiments were performed in a group of participants whereas simulations were performed based on a single musculoskeletal model with fixed anthropometry rather than a set of virtual participants with varying body parameters [57]. We chose this approach for computational reasons and because differences between simulations and average experimental data well exceeded inter-subject differences in experimental data after normalization. Overall, it is very likely that the effect of errors in the musculoskeletal models, metabolic energy models, and cost function were more important than the errors in representing the task given the magnitude of differences between experimental and simulated metabolic power.

When using physics-based simulations to predict the effect of an intervention it is crucial to know the accuracy of the simulated walking kinematics, kinetics, and energetics. In some cases, it might be helpful to be able to predict the direction of the change even if the predicted magnitude is not accurate. For example, when designing assistive devices, knowing whether a design choice will increase or decrease step frequency or energy consumption might be sufficient. In this study, for the range of walking conditions tested, the explained variance in stride frequency and metabolic cost was generally high (Figs 1 and 2), suggesting that the model correctly captured the direction of the intervention's effects. An important future step is to assess whether simulations can be used for designing better interventions. The simulations presented here rely on a predefined musculoskeletal model and we altered the task constraints. When using simulations for designing optimal interventions, e.g., to determine the stiffness of an orthoses, model parameters become optimization variables. It is unclear whether the accuracy of capturing the effect of different task constraints and different model parameters is comparable. For example, a lower-limb exoskeleton that replaces the entire ankle, knee, and hip torque would be optimal in our simulations. This is however not observed in experimental studies where submaximal assistance is optimal [58,59]. This demonstrates that other modelling assumptions (e.g., rigid contact between exoskeleton and human, influence on balance, …) might become critical in designs that have a large impact on the simulated walking motion and/or muscle activity.

In this study, we predicted various walking conditions without explicitly (i.e., part of cost function or as constraints) or implicitly using validation data to tune simulation parameters, apart from walking at 1.3 m/s [10]. This systematic approach revealed substantial limitations, particularly in the metabolic energy model and the mechanical work done by muscle fibers. We believe that this benchmarking process should be iterative, preferably incorporating novel data when adapting models or model parameters in the simulation (i.e., weights in cost function, constraints on muscle coordination). This approach would minimize the risk of overfitting the model to a specific walking condition and enhance the model's ability to generalize across various walking conditions.

## Conclusion

This study demonstrates that while model-based simulations can reasonably predict variations in human walking kinematics and spatio-temporal parameters, substantial limitations exist in the prediction of metabolic energy. These inaccuracies likely stem from two issues. First, the predicted positive muscle fiber work is unrealistically high, which may be caused by assumptions and errors in the musculoskeletal model including its interacting with the environment and/or its many parameters. Second, the phenomenological metabolic energy models based on Hill-model states and inputs result in unrealistically high efficiencies [18]. This highlights the need for more accurate musculoskeletal and energetics models to improve the predictive capability of human movement simulations. Addressing these limitations through improved modeling and iterative validation will enhance the reliability of simulations, especially in predicting metabolic power during walking. Ultimately, refining these models will be crucial for enabling the use of simulations for the design of assistive devices or for optimal treatment selection.

## Supporting information

**S1 File. Supplementary files.**
(PDF)

**S1 Video. Predicted walking motions in various gait conditions.**
(MP4)

## Acknowledgments

We want to thank all the authors that published their raw data as a supplement or dataset. Without this, or the use of tools like AddBiomechanics, benchmarking predictive simulations would be even more time intensive. We also want to thank Koen Lemaire, and Tim van der Zee for the insightful discussion on metabolic energy models; Lars D'Hondt for the numerous developments in the predictive simulation code; Keenon Werling for the development and help with addBiomechanics; Steve Collins, Katie Poggensee and Antoine Falisse for the discussions on simulations with walking with an ankle-foot exoskeleton. The (lack of) validity of these simulations was the basis of this research project.

## Author contributions

**Conceptualization:** Maarten Afschrift, Dinant Kistemaker, Maarten Bobbert, Friedl De Groote.

**Data curation:** Maarten Afschrift, Friedl De Groote.

**Formal analysis:** Maarten Afschrift, Dinant Kistemaker, Maarten Bobbert, Friedl De Groote.

**Funding acquisition:** Maarten Afschrift, Friedl De Groote.

**Investigation:** Maarten Afschrift, Dinant Kistemaker, Maarten Bobbert, Friedl De Groote.

**Methodology:** Maarten Afschrift, Dinant Kistemaker, Maarten Bobbert, Friedl De Groote.

**Project administration:** Maarten Afschrift, Friedl De Groote.

**Resources:** Maarten Afschrift, Friedl De Groote.

**Software:** Maarten Afschrift.

**Supervision:** Dinant Kistemaker, Friedl De Groote.

**Validation:** Maarten Afschrift, Friedl De Groote.

**Visualization:** Maarten Afschrift, Friedl De Groote.

**Writing – original draft:** Maarten Afschrift, Dinant Kistemaker, Maarten Bobbert, Friedl De Groote.

**Writing – review & editing:** Maarten Afschrift, Dinant Kistemaker, Maarten Bobbert, Friedl De Groote.

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
