## [Decision Letter · Decision Letter 0]

10 Apr 2025

Benchmarking the predictive capability of human gait simulations.

PLOS Computational Biology

Dear Dr. Afschrift,

Thank you for submitting your manuscript to PLOS Computational Biology. After careful consideration, we feel that it has merit but does not fully meet PLOS Computational Biology's publication criteria as it currently stands. Therefore, we invite you to submit a revised version of the manuscript that addresses the points raised during the review process.

Please submit your revised manuscript within 60 days Jun 10 2025 11:59PM. If you will need more time than this to complete your revisions, please reply to this message or contact the journal office at ploscompbiol@plos.org. Please include the following items when submitting your revised manuscript:

We look forward to receiving your revised manuscript.

Kind regards,

Fabian Spill

Academic Editor

PLOS Computational Biology

Pedro Mendes

Section Editor

PLOS Computational Biology

**Journal Requirements:**

At this stage, the following Authors/Authors require contributions: Maarten Afschrift, Dinant Kistemaker, and Friedl De Groote. Please ensure that the full contributions of each author are acknowledged in the "Add/Edit/Remove Authors" section of our submission form.

2) Please amend your detailed Financial Disclosure statement. This is published with the article. It must therefore be completed in full sentences and contain the exact wording you wish to be published.

1) State what role the funders took in the study. If the funders had no role in your study, please state: "The funders had no role in study design, data collection and analysis, decision to publish, or preparation of the manuscript.".

**Reviewers' comments:**

Reviewer's Responses to Questions

**Comments to the Authors:**

**Please note that one of the reviews is uploaded as an attachment.**

Reviewer #1: This manuscript benchmarks an existing computational method for predicting human gait. The prediction method is used to replicate several (previously published) experiments where humans were exposed to changes in task or environment, and their changes in gait and energy consumption were measured. Points of agreement and disagreement between predictions and experiments were noted and discussed.

This work is important and very much needed. The ability to predict human gait adaptations offers enormous promise for simulation-based design for a broad range of assistive technologies and rehabilitation strategies. Although prediction methods exist, claims of success have not been sufficiently convincing. The benchmark tests presented in this paper are excellent and represent a first attempt to systematically find the weaknesses of prediction methods. It is only though this type of work that gait prediction methods can be improved to become sufficiently trusted for practical applications.

The manuscript is well written and was a pleasure to read. The methods are sound and executed competently. The discussion is mostly complete. All data and models are shared through a GitHub repository. This is a big deal, because it gives others the opportunity to replicate the work and possibly improve on it. My specific comments (below) are intended to help improve the presentation and increase the impact of the work.

1. Line 41. “kinematic” should possibly be “kinematics”.

2. Line 55: In addition to better models of muscle mechanics and energetics, and passive mechanics, better optimality criteria may also be needed. See also below for a more detailed suggestion for the Discussion.

3. Line 156-157. Cost function weights were hand-tuned to obtain good predictions for a single walking condition. This is somewhat contradictory to what was said in the previous paragraph, that such an approach can lead to overfitting, and produce poor predictions for cases outside of the “training” data. While I agree with this approach, scientifically (you benchmark an existing method), it’s good to keep in mind that the cost function may not have been the best one. This can be addressed in the Discussion. See a later comment.

4. Line 159. The term “linear muscle actuators” is not familiar to me. Please reword or clarify with a reference.

5. Equation 1: for clarity, please right-justify the equation number so it is clearly separate from the equation itself.

6. Equation 1: The notation is potentially confusing to someone who is not familiar with previous work. You could either explicitly do summations over muscles, degrees of freedom, etc., or use the notation from Falisse et al. (2019) to show that each term is actually a squared vector norm over multiple components.

7. Line 200, “We used the weights (w1- w5) as published in (21).” The reference (21) does not provide values for the weights, perhaps (10) should be used as a reference. Also, it would be helpful to list the weights in a Table (in Supplement). When providing numerical values for the weights, it is important to specify the units of the variables they are applied to, i.e. energy rates must be in Watts, generalized accelerations in m/s^2 or rad/s^2, etc. Otherwise the weight values are not meaningful.

8. Lines 330-331. “The observed increase in hip flexion moment and extension moment with faster walking was predicted accurately”. However, Figure S4 shows that the horizontal ground reaction force (in both horizontal directions) is severely underestimated. It seems unlikely that the hip moment is predicted correctly when the horizontal GRF is not. It may be that there are also prediction errors in vertical GRF and/or kinematics, in such a way that the total effect of these errors on hip moment is largely canceled out. The error in horizontal GRF should be mentioned and discussed, because it suggests that the model does not walk like the human participants. In fact, Figure S4 is never mentioned in the manuscript at all.

9. Lines 396-397. “The simulation did not account for the total activated muscle volume, which could have influenced the results (Figure 3 G).” Please clarify what is meant by “did not account for”.

10. Figure 3G. The Figure says “muscle mass”, while the text says “muscle volume”, so it is not clear what it actually is. It would help to explain how this variable is calculated, and place the appropriate unit (kg or m^3) on the vertical axis.

11. This also reminds me of an important paper by Holmberg and Klarbring (2012) which proved that muscle activations (or squared activations) should not be added up in an optimization objective without weighting them by factors that make the objective invariant to anatomical discretization. Muscle volume, or mass, or PCSA, would all be appropriate weighting factors. Sometimes, in 3D models, a muscle is divided into many components, each with their own line of action. Such muscles then get too much weight in an activation-based optimization objective, unless their contributions are weighted appropriately. This may be good to mention in the Discussion, in the paragraph about the optimization objective.

12. Line 492. “The underestimation of metabolic power”. In fact, the model overestimated the metabolic power! It is only the change in metabolic power that is underestimated. Please make this distinction. It is not clear to me how an unrealistically high muscle efficiency can cause the metabolic power to be overestimated. Nevertheless, the changes were underestimated. Could it be that the metabolic power had too much weight in the optimization objective, causing gait adaptations to avoid increasing it?

13. Line 501. Similar to the previous comment, “underestimation of metabolic cost”, should be “underestimation of changes in metabolic cost”.

14. Lines 578-580. These two sentences state an obvious but important fact, but then the topic is changed without any discussion of the optimization objective. This should be significantly expanded. The optimization objective is directly taken from Falisse et al. (2019). Different weights, or different objectives altogether, could be considered. In a previous comment, I already mentioned that the activation objective is not properly weighted. Furthermore, others (Ackermann 2010, Rasmussen 2001) have suggested fatigue-like criteria such as minimizing only the largest muscle activation (which then is no longer subject to Holmberg and Klarbring’s concerns). Also, the squaring of each muscle’s energy rate before adding and integrating certainly does not result in anything resembling the cost of transport. I know that Falisse tried without squaring and did not like the predictions, but physiologically, this objective makes no sense. Also, the cost function terms related to energy, activation, and generalized accelerations are probably highly correlated. It may well be that Falisse could have obtained equally good predictions by weighting them differently. In that case, there was overfitting, which could explain some of the prediction errors in this important benchmark study. I believe there is much work to do on cost functions, and testing them with benchmark tests exactly as presented in this manuscript. It is scientifically acceptable to use just one (previously published) cost function in this paper, but it is not acceptable to discuss the results without a robust discussion of possible problems with the cost function. The reader deserves more than just stating the obvious fact that the cost function influenced the predictions.

15. To help explain some incorrect predictions, and give guidance for future research, it might be helpful to have some plots of the five (weighted) cost function components as a function of the test condition (speed, added weight, etc.). Then we might see if any component is dominating, and how each of them might be driving the gait adaptations. This could be in the Supplement and be cited in the discussion of the optimization objective.

16. I examined the code repository, and it was well designed and well organized. The code is sufficiently readable for readers who are interested in replicating or improving the work. However, I would recommend testing by an external person, or on a computer that has nothing installed yet, to find the problems that readers will encounter. I encountered three problems before giving up. Yes, I could have probably made it work, but I would be happier and more likely to engage with the code if it just works. The code is excellent and I would hate to see people giving up on it because of problems that could have been prevented.

Problem 1: The instructions say to add the repository to the Matlab path. Adding the root folder would not be sufficient, so I added it with all subfolders which is overkill. An install script could take care of setting the path properly.

Problem 2: I tried running ConvertModelsKoelewijn.m and it crashed because the repository path was hardcoded as C:\users\mat950\Documents\... which obviously will work only on the author’s computer. This was easy to fix, but it would be better to use mfilename('fullpath') to automatically determine what the repository path is on the user’s computer.

Problem 3: After fixing the repository path, the script crashed when it could not find the file ‘…\PredSim_gait_conditions\PredSim\Subjects\Falisse2022_8pDecline\F_Falisse2022_8pDecline_IO.mat'. I did not try to debug this.

Reviewer #2: Please see attachment

**Have the authors made all data and (if applicable) computational code underlying the findings in their manuscript fully available?**

Reviewer #1: Yes

Reviewer #2: Yes

PLOS authors have the option to publish the peer review history of their article (what does this mean? ). If published, this will include your full peer review and any attached files.

**Do you want your identity to be public for this peer review?** For information about this choice, including consent withdrawal, please see our Privacy Policy .

Reviewer #1: **Yes: ** Antonie J. (Ton) van den Bogert

Reviewer #2: No

**Figure resubmission:**

**Reproducibility:**



---

## [Decision Letter · Decision Letter 1]

20 Oct 2025

PCOMPBIOL-D-24-02149R1

Benchmarking the predictive capability of human gait simulations.

PLOS Computational Biology

Dear Dr. Afschrift,

Thank you for submitting your manuscript to PLOS Computational Biology. The reviewers have just minor comments, please submit a revised version of the manuscript that addresses the points raised during the review process.

Please submit your revised manuscript within 30 days Dec 20 2025 11:59PM. If you will need more time than this to complete your revisions, please reply to this message or contact the journal office at ploscompbiol@plos.org. Please include the following items when submitting your revised manuscript:

We look forward to receiving your revised manuscript.

Kind regards,

Fabian Spill

Academic Editor

PLOS Computational Biology

Pedro Mendes

Section Editor

PLOS Computational Biology

**Reviewers' comments:**

Reviewer's Responses to Questions

**Comments to the Authors:**

Reviewer #1: I am pleased with the revision, and congratulate the authors on publishing this work. It will have substantial impact by providing a conceptual foundation, code, and data for further research into the validity and identification of optimization objectives for human movement.

I did catch a couple of typing errors.

(referring to line numbers in the manuscript where revisions are marked)

Line 96, "principals" should be "principles".

Line 769, "waking" should be "walking".

Reviewer #2: I would like to thank the authors for the thorough revision that was performed. I just have a few comments left:

Main comment (paragraph from line 681 onwards): I want to point the authors to an alternative explanation for the unrealistic kinematics and kinetics that could also be mentioned in addition: when using it as an objective, the optimization favors using lengthening work because of its high efficiency, which even makes some of the positive work free in a way. This to me is different to energy optimization as it happens by humans and by other metabolic models.

Line 611: I find the phrasing "It is most likely a good idea" not strong enough, because this is something that should be done. Consider rephrasing.

Line 705: e.g.  add a comma after

**Have the authors made all data and (if applicable) computational code underlying the findings in their manuscript fully available?**

Reviewer #1: Yes

Reviewer #2: None

PLOS authors have the option to publish the peer review history of their article (what does this mean? ). If published, this will include your full peer review and any attached files.

**Do you want your identity to be public for this peer review?** For information about this choice, including consent withdrawal, please see our Privacy Policy .

Reviewer #1: **Yes: ** Antonie J. (Ton) van den Bogert

Reviewer #2: No

**Figure resubmission:**
---

## [Editor Report · Decision Letter 2]

23 Oct 2025

Dear Dr. Afschrift,

We are pleased to inform you that your manuscript 'Benchmarking the predictive capability of human gait simulations.' has been provisionally accepted for publication in PLOS Computational Biology.

Best regards,

Fabian Spill

Academic Editor

PLOS Computational Biology

Pedro Mendes

Section Editor

PLOS Computational Biology

---

## [Editor Report · Acceptance letter]

PCOMPBIOL-D-24-02149R2

Benchmarking the predictive capability of human gait simulations.

Dear Dr Afschrift,

I am pleased to inform you that your manuscript has been formally accepted for publication in PLOS Computational Biology. Your manuscript is now with our production department and you will be notified of the publication date in due course.

With kind regards,

Judit Kozma
